# Prenylquinones in Human Parasitic Protozoa: Biosynthesis, Physiological Functions, and Potential as Chemotherapeutic Targets

**DOI:** 10.3390/molecules24203721

**Published:** 2019-10-16

**Authors:** Ignasi B. Verdaguer, Camila A. Zafra, Marcell Crispim, Rodrigo A.C. Sussmann, Emília A. Kimura, Alejandro M. Katzin

**Affiliations:** 1Department of Parasitology, Institute of Biomedical Sciences, University of São Paulo, São Paulo 05508000, Brazil; ig_la123@hotmail.com (I.B.V.); czafra@javeriana.edu.co (C.A.Z.); marcellcrispim@gmail.com (M.C.); eakimura@usp.br (E.A.K.); 2Centro de Formação em Ciências Ambientais, Universidade Federal do Sul da Bahia, Porto Seguro 45810-000 Bahia, Brazil

**Keywords:** drug targets, parasitic protozoa, prenylquinones, phylogeny, neglected diseases

## Abstract

Human parasitic protozoa cause a large number of diseases worldwide and, for some of these diseases, there are no effective treatments to date, and drug resistance has been observed. For these reasons, the discovery of new etiological treatments is necessary. In this sense, parasitic metabolic pathways that are absent in vertebrate hosts would be interesting research candidates for the identification of new drug targets. Most likely due to the protozoa variability, uncertain phylogenetic origin, endosymbiotic events, and evolutionary pressure for adaptation to adverse environments, a surprising variety of prenylquinones can be found within these organisms. These compounds are involved in essential metabolic reactions in organisms, for example, prevention of lipoperoxidation, participation in the mitochondrial respiratory chain or as enzymatic cofactors. This review will describe several prenylquinones that have been previously characterized in human pathogenic protozoa. Among all existing prenylquinones, this review is focused on ubiquinone, menaquinone, tocopherols, chlorobiumquinone, and thermoplasmaquinone. This review will also discuss the biosynthesis of prenylquinones, starting from the isoprenic side chains to the aromatic head group precursors. The isoprenic side chain biosynthesis maybe come from mevalonate or non-mevalonate pathways as well as leucine dependent pathways for isoprenoid biosynthesis. Finally, the isoprenic chains elongation and prenylquinone aromatic precursors origins from amino acid degradation or the shikimate pathway is reviewed. The phylogenetic distribution and what is known about the biological functions of these compounds among species will be described, as will the therapeutic strategies associated with prenylquinone metabolism in protozoan parasites.

## 1. Introduction

### 1.1. Public Health Relevance of Parasitic Protozoa

Among parasites, protozoa are known to be etiological agents of important diseases such as Chagas disease (*Trypanosoma cruzi*), African human trypanosomiasis (*Trypanosoma brucei*), severe leishmaniasis (*Leishmania* spp.), malaria (*Plasmodium* spp.), and severe coccidiosis [1,2]. In general, most neglected diseases occur frequently in tropical and subtropical countries, where these diseases are estimated to affect more than one billion people per year [3,4]. For example, in 2016, more than 216 million people were estimated to have been affected by malaria, approximately 445,000 of whom died. Another example is human leishmaniasis, which is caused by the parasite *Leishmania* spp. and also widely distributed in Africa, Asia, and America [5,6]. Leishmaniasis causes approximately 70,000 deaths and 2 million infections per year [5,6]. Unfortunately, drug resistance for some treatments is emerging, generally against drugs that have traditionally been used to treat the abovementioned diseases [7]. Therefore, an improved understanding of the factors that influence immunity and discovery of new etiological treatments for public health interventions are urgently required [4].

Because of the parasitic life cycle and transmission process, parasites are usually able to live under diverse and occasionally extreme environmental conditions. For example, monoaxenic protozoans, such as *Giardia* spp. or *Cryptosporidium* spp., which are able to survive in the vertebrate digestive system as well as in host-free environments based on resistance stage [8]. Other parasites, such as *Trypanosoma* spp., *Leishmania* spp., and *Plasmodium* spp., possess heteroxenic cycles and, thus, are able to survive in different organisms [8,9,10]. Protozoan parasites are subjected to extreme environmental changes and stress, including nutrient depletion, oxygen saturation, extreme temperatures, and oxidative environments [11,12,13,14]. For example, *Plasmodium* and *Haemoproteus* parasites have developed multiple antioxidant mechanisms, including heme polymerization [15]. Similarly, *Plasmodium* spp., *Leishmania* spp., and *Trichomonas vaginalis* possess superoxide dismutase or glutathione, among others antioxidant systems [13,16,17]. *Trypanosoma brucei* is also subjected to periods of nutritional stress in the invertebrate host, and it is known that *Giardia intestinalis* is extremely well adapted for survival in environments with low oxygen saturation [18,19]. Interestingly, these adaptive mechanisms include genes and metabolic pathways from, typically, bacteria, eukaryotic heterotrophs, and photosynthetic organisms. These genes are present due to the varied phylogenetic origin of protozoa as well as the different endosymbiotic processes that have occurred throughout the evolutionary history of these parasites [20,21,22]. Moreover, a brief description of the impact of protozoan parasite diseases on world health is summarized in Appendix A.

### 1.2. Endosymbiotic Events of Parasitic Protozoa

Protozoa were first grouped as a single monophyletic group and were considered ancestors of the animal kingdom [23]. However, molecular studies showed that protozoa are not a single monophyletic group [23]. Eventually, the ancient subkingdom Protozoa was rejected and reclassified with several algae and fungi [24]. The classification includes several phyla, including Sarcomastigophora, Apicomplexa, Ciliophora, Labyrinthomorpha, Macrospora, Ascetospora, and Myxospora [23]. However, only Sarcomastigophora, Ciliophora, and Apicomplexa contain major human pathogens [9,23,24]. Sarcomastigophora includes important parasitic families such as Endamoebidae (including *Entamoeba histolytica*), Trypanosomatidae (*Trypanosoma* spp., *Leishmania* spp.), Hexamitidae (*Giardia* spp.), and Trichomonadidae (*T. vaginalis*) [25]. The phylum *Apicomplexa* includes parasitic families such as Eimeriidae (*Isospora* spp.), Cryptosporidium (*Cryptosporidium* spp.), Sarcocystidae (*Sarcocystis* spp. and *Toxoplasma gondii*) and Plasmodiidae (*Plasmodium* spp.). Finally, the Ciliophora phylum contains the Balantidiidae family, which is mainly represented by *Balantidium coli*. Other classifications include apicomplexan parasites together with dinoflagellates and ciliates within the alveolates group [24,26]. There is evidence indicating that during evolution, alveolate ancestors underwent two endosymbiotic processes. Primary endosymbiosis occurred between a phagotroph ancestor of the current algae and a cyanobacterium [26,27] and originated in the Archaeplastida group, which includes Glaucocystophytes, *Rhodophyta* (red algae), and Viridiplantae (green algae and plants). Secondary endosymbiosis probably occurred between a second alveolate phagotroph ancestor and red algae [21]. Due to these phenomena, several species of photosynthesizing dinoflagellates that produce chlorophyll and peridinin [27].

Several apicomplexans still harbor a non-photosynthetically modified chloroplast called the apicoplast. The apicoplasts of *Plasmodium falciparum* and *T. gondii* seem to be strongly associated with mitochondria [28,29], and both organelles share metabolic pathways such as heme biosynthesis in *Plasmodium* [29,30]. Due to the fact of these endosymbiotic processes, apicomplexan parasites possess metabolic pathways that are typically present in photosynthetic organisms or bacteria. Some examples are amylopectin biosynthesis [31], aromatic ring biosynthesis [32], the calcium-dependent protein kinase (CDPK) multigene family [33] and isoprenoid biosynthesis by the methyl erythritol phosphate (MEP) pathway [34]. Due to the presence of these pathways, apicomplexan parasites are sensitive to some herbicides, as described in this review [35,36]. Recent studies also indicate that trypanosomatids share a common ancestor with euglenids, a group containing photosynthetic organisms [37]. Some authors suggested that this plastid was recently acquired by the photosynthetic euglenid common ancestor [37,38] while other researchers suggests that the plastids appeared at the beginning of euglenozoic evolution before the bodonids and euglenids diverged [37,38]. After this event, bodonids may have lost the plastid and subsequently diversified to several phylogenetic groups, including trypanosomatids. Finally, archezoan protists (including diplomonads, microsporidia, and trichomonads) were thought to have separated from eukaryotic organisms before the endosymbiotic event, which originated the mitochondria [38,39,40]. However, molecular studies revealed that the *T. vaginalis* genome encodes mitochondrial proteins such as a chaperone protein (HSP70) associated with a sister group of α-purple bacteria and proteobacteria [40]. Similarly, mitochondrial enzymes such as pyridine nucleotide transhydrogenase and the chaperonin cpn60, associated with mitochondrial linage sequences, were identified in *E. histolytica* and *Giardia lamblia* [38]. These facts suggested that an endosymbiosis event with a progenitor of mitochondria occurred early in the evolutionary history of archezoan protists [38,40].

Due to the diversified phylogenetic origin and the horizontal gene transfer events previously discussed, there exist very phylogenetically distant and diversified metabolic pathways among protozoan organisms [20,21,22,37,39]. As previously mentioned, some substances, such as prenylquinones (Figure 1), have a heterogeneous distribution in nature due to the diffusion of these molecules between different taxa via endosymbiotic processes.

## 2. Prenylquinones and the Study of These Compounds

As the name implies, prenylquinones are structurally composed of an aromatic naphthoquinones or benzoquinones attached to an isoprene chain (Figure 1). The classic nomenclature classifies these substances based on their naphthoquinone/benzoquinone group (example: ubiquinone, UQ; menaquinone, MQ) and numerically indicates the number of side-chain isoprene units (UQ-n, UQn). Prenylquinones are apolar compounds that have the ability to protonate and deprotonate and, thus, most prenylquinones are involved in electron transfer reactions in the cell lipid phase. Many prenylquinones are commonly known as vitamins because these compounds are necessary components of animal dietary intake [41,42,43,44]. It seems that the first prenylquinones were MQs, which appeared in the context of a primitive atmosphere that lacked oxygen [45]. Changes in the Earth’s gaseous composition led to the appearance of UQs among other prenylquinones [45]. Currently, many different prenylquinones exist in nature that, because of their chemical characteristics, are capable of developing specific activities in each organism. Among others, the prenylquinone group includes UQs, plastoquinones (PQs), plastocromanol, MQs, phylloquinone (PK), tocopherol, rhodoquinones, chlorobiumquinone (ChQ), and thermoplasmaquinone (TpQ) [45]. Each substance presents a different polarity and redox potential, which determines the cellular location of the compound and the ability of the compound to protonate and deprotonate at different oxygen saturations [45,46]. The isoprenic chain size has also been shown to influence catalytic activity, activation energy, and kinetics among other chemical characteristics [46].

Even in the same organism, the biosynthesis and proportions of different prenylquinones can be modified as a result of environmental changes. For example, *Escherichia coli* modifies the UQ–MQ pool ratio as a function of environmental oxygen saturation, and *Euglena gracilis* also modifies rhodoquinone and UQ biosynthesis for similar purposes [47,48].

Among other functions, prenylquinones are involved in lipoperoxidation defense, mitochondrial respiration and photosynthesis and act as enzymatic cofactors [45]. For example, PQs in plants and cyanobacteria are involved in photosynthetic electronic transport (PSII) and cell respiration and participate in antioxidant defense, similar to tocopherol [49]. Ubiquinone is described to mainly participate in mitochondrial electronic transport between complex I or II and complex III in eukaryotes [50]. However, UQ can also be involved in photosynthesis in some organisms, such as *Rhodospirillum rubrum* [45]. Similarly, MQs are mainly involved in photosynthesis and respiration processes in prokaryotes [45,51]. The existence of trans-plasma membrane electron transport (TPMET) chains between extracellular and intracellular compounds in several organisms is known. Some electron carriers identified in TPMET are NADH, glutathione, and several prenylquinones, such as tocopherol, UQ, PK, and MQ [52,53]. Among other factors, TPMET is essential for maintenance of internal homeostasis and prevention of oxidative stress as well as ferric ion reduction for intracellular incorporation [52]. There are specialized enzymes involved in these types of reactions, such as NAD(P)H reductase, which reduces UQ to its semiquinone form, which in turn reduces tocopherol [52]. Subsequently, tocopherol reduces extracellular ascorbic acid [52].

Several researchers have focused on the identification of new prenylquinones in different organisms [41,45,54]. The goal of these studies has been to better understand the phylogeny, bioenergetics and electron transfer metabolism of the studied organisms. In pathogenic organisms, the functions and biosynthetic pathways of prenylquinones are interesting drug targets. Aurachin is an MQ biosynthesis inhibitor that has been previously investigated for antimycobacterial activity [55]. Another example is the *Ascaris suum* NADH–rhodoquinone reductase and rhodoquinol–fumarate reductase, both of which are considered excellent enzymatic drug targets, and specific inhibitors of these enzymes, such as nafuredin, have already demonstrated promising anthelmintic activity [56]. Ultimately, a summary about prenylquinone functions is available in Table 1.

## 3. Biosynthesis of Aromatic and Isoprenic Precursors of Prenylquinones in Parasitic Protozoa

As previously discussed, most natural occurring prenylquinones are substances that are structurally composed of a naphthoquinone- or benzoquinone-modified aromatic head group bound to an isoprenic side chain. Therefore, the biosynthesis of all prenylquinones involves the biosynthesis of both portions’ precursor. However, the aromatic head group and isoprenic side chain can be formed in different ways depending on the protozoan. The biosynthetic pathways of the aromatic head group precursors and the lateral chain have been studied as drug targets and are discussed below. The specific steps for the biosynthesis of each prenylquinone are discussed in Section 4 and the information on the distribution of the pathways in pathogenic protozoa is included in Table 2.

### 3.1. Aromatic Head Group Biosynthesis

The shikimate pathway, together with specific amino acid degradation, supplies the precursors required for the biosynthesis of a wide variety of compounds (folates, amino acids and prenylquinones, among others). The presence of these metabolic pathways in parasitic protozoa have been briefly discussed (Figure 2).

#### 3.1.1. Amino Acid Degradation

##### Generalities about Amino Acid Degradation

There are several routes for the conversion of the aromatic amino acids tyrosine and phenylalanine to 4-hydroxybenzoate, 4-hydroxyphenylpyruvate (4HPP) and chorismate, which are required for the biosynthesis of all prenylquinones and other aromatic compounds. However, these pathways remain poorly studied in parasitic protozoa. Tyrosine degradation begins with nitrogen transfer from the aromatic amino acid to alpha ketoglutarate, generating glutamate and 4HPP. This reaction can be catalyzed by several aminotransferases, including tyrosine aminotransferase (TAT). Additionally, phenylalanine can be transformed to prephenate by the action of prephenate dehydratase and consecutively to 4HPP by prephenate dehydrogenase [62,63,64,88,89]. The enzyme 4-hydroxyphenylpyruvate dehydrogenase (HPPD) can convert 4HPP to homogentisate, the aromatic precursor of tocopherol and PQ in plants [45].

There exists another pathway for the conversion of amino acids to 4-hydroxybenzoate. This pathway starts with an oxidative deamination, catalyzed by the flavoenzyme L-amino acid oxidase, which converts tyrosine or other amino acids to the corresponding ketoacid while generating ammonia and hydrogen peroxide [90,91]. In the second step of the pathway, the formation of 4-hydroxyphenylactate (HPL) occurs via the action of a NAD^+^-dependent HPP reductase that has been previously studied in several plant models [92]. In the third step of the pathway, HPL is converted into 4-coumarate by a mechanism not has been elucidated to date [93]. Then, in the fourth step of the pathway, 4-coumarate, ATP, and coenzyme A (CoA) react as substrates of the enzyme 4-coumarate-CoA ligase, generating AMP, diphosphate and 4-coumaroyl-CoA [82]. Loscher et al. suggested that the fifth step can be performed by beta oxidation, in which 4-coumaroyl-CoA would be oxidatively decarboxylated to 4-hydroxybenzoyl-CoA at the expense of a CoA. In the sixth step of this enzymatic pathway, 4-hydroxybenzoyl-CoA derived from the previous reaction is hydrolyzed to a thioester group, generating 4-hydroxybenzoate [82].

##### Amino Acid Degradation in Parasitic Protozoa

Conversion of the aromatic amino acids tyrosine and phenylalanine to 4-hydroxybenzoate, 4-hydroxyphenylpyruvate (4HPP), and chorismate remains poorly studied in parasitic protozoa. Only tyrosine degradation has been demonstrated to be essential in *Leishmania* and this pathway seems to be an excellent drug target [63]. HPPD and TAT inhibitors have been shown to compromise tyrosine degradation and tocopherol biosynthesis in *Leishmania* and *P. falciparum* [64,90,91]. Alternatively, prephanate can also be converted to chorismate by the chorismate mutase enzyme. Subsequently, chorismate can be converted to 4-hydroxybenzoate by the action of chorismate-pyruvate lyase, which is then used for vitamin K, UQ or PQ biosynthesis depending on the organism [45].

#### 3.1.2. Shikimate Pathway

##### Generalities about Shikimate Pathway

The shikimate pathway is another source of chorismate in not only photosynthetic organisms such as algae and plants but also in some bacteria and apicomplexan parasites [73,83]. The shikimate pathway begins when the enzyme 3-deoxy-arabinoheptulosonate-7-phosphate (DAHP) synthase forms 3-deoxy-arabinoheptulosonate-7-phosphate by condensing phosphoenolpyruvate (PEP) and d-erythrose-4-P. 3-deoxy-arabinoheptulosonate-7-phosphate is reduced and dehydrated, forming shikimic acid. This molecule is phosphorylated and then reacts again with PEP to form 5-*O*-(1-carboxyvinyl)-3-P-shikimate by the action of the enzyme 5-enolpyruvylshikimate-3-phosphate synthase. The herbicide glyphosate is inhibited by this enzyme [68]. Finally, chorismite is formed as a consequence of the phosphorylation of 5-*O*-(1-carboxyvinyl)-3-*P*-shikimate by the enzyme chorismite synthase. Chorismate can follow different pathways depending on the product to be synthesized.

##### Shikimate Pathway in Parasitic Protozoa

The shikimate pathway appears to be active in the apicomplexan parasite cytosol and is thought to be essential for the biosynthesis of folates, prenylquinones and some amino acids. It is not surprising that the herbicide glyphosate and shikimic acid analogs have previously exhibited antiparasitic activity against apicomplexan pathogens [74,75,94]. However, Choudhary et al. [75] demonstrated *P. berghei* mutants lacking the enzyme chorismate synthase were viable in the mammalian host and the mosquito vector. Therefore, Choudhary et al. suggested the shikimate pathway not to be essential for the parasite viability and consequently not a good drug target [75].

### 3.2. Isoprenic Side Chain Biosynthesis

Isoprenoid biosynthesis occurs in almost all known living organisms [45]. In addition to prenylquinones, isoprenoids and isoprenylated compounds include diverse and important molecules such as isoprenylated proteins, sterols, isoprene compounds bound to RNA, dolichols and carotenoids [95]. There are two well-characterized pathways for isoprenoid biosynthesis: the mevalonate (MVA) pathway and the MEP pathway (Figure 3). The final result of both pathways is the formation of isopentenyl pyrophosphate (IPP) or its isomer dimethylallyl pyrophosphate (DMAPP), both of which are considered to be simple isoprenic units required for the formation of differently sized chains [45,95]. For both pathways, well-known specific inhibitors exist, such as fosmidomycin for the MEP pathway and statins for the MVA pathway. Most archaeal species use the MVA pathway, while most bacteria use the MEP pathway [96,97,98,99]. However, there are several exceptions. For example, *Myxococcus fulvus* uses the MVA pathway, and *Streptomyces* spp. have both isoprenic biosynthetic pathways [96]. The MVA pathway is also used as the only source of isoprenic precursors by all known animals and fungi [97]. Higher plants have the two pathways compartmentalized in different subcellular spaces, and these pathways are used to provide isoprenic precursors for different biosynthetic processes. The MVA pathway is found in the cytosol for UQ biosynthesis among other compounds, and the MEP pathway is located in the chloroplast, providing isoprenic precursors for chlorophyll, PK or PQ. In algae and protists, there are a few pathways for biosynthesis of isoprenoids [68,97,98,100,101].

In *Leishmania* and *Trypanosoma* species, indicators of leucine dependent pathways were observed for isoprenoid biosynthesis [70,71,102]. These pathways can be independent of the MVA pathway or coupled to the MVA pathway with or without conversion of leucine to acetyl-CoA for 3-hydroxy-3-methyl-glutaryl-CoA (HMG-CoA) regeneration [70,71,102]. In the case of the MVA-coupled pathway without the conversion of leucine to acetyl-CoA, leucine can be reduced to HMG-CoA in *T. brucei*. Alternatively, the MVA pathway-independent incorporation of leucine into isoprenoids, an intermediate step in leucine breakdown (probably to dimethylcrotonyl-CoA), could be directly reduced to produce isopentenol in *Leishmania mexicana* [70]. The incorporation of leucine to isoprenic chains without the conversion of leucine to acetyl-CoA has also been demonstrated in other widely divergent organisms, such as halophilic archaea [78].

In *P. falciparum*, some authors suggested that alternative MVA/MEP pathways could be active. Cassera et al. [103] found intermediates of the MEP pathway in *P. falciparum* but also observed a poor inhibitory effect of fosmidomycin on some isoprenylated final products in specific intraerythrocytic stages. Cassera et al. suggested that an as-yet-unknown isoprenic source could be present in the parasite [103]. The author cited a hypothetical pathway initiated by pentose phosphate cycle intermediates, for the existence of which there is only some evidence in cyanobacteria [104]. However, there remain many uncertainties associated with pentose phosphate cycle intermediates/leucine-dependent pathways that are not coupled to the MVA pathway as an isoprenic precursor source. For these reasons, the sections below focus on the MVA and MEP pathways.

#### 3.2.1. Mevalonate Pathway

##### Generalities about the Mevalonate Pathway

The MVA pathway was the first isoprenoid synthesis pathway to be discovered and for many years was believed to be the only isoprenoid source in plants, fungi, archaebacteria, eubacteria, and some protozoa groups [45,96,99]. The MVA pathway begins with the condensation of two acetyl-CoA molecules for HMG formation) formation, a reaction catalyzed by the enzyme HMG-CoA synthase. Then, the enzyme HMG-CoA reductase catalyzes mevalonate formation, a reaction that can be inhibited by statins [97]. Finally, mevalonate is phosphorylated twice and decarboxylated to IPP [105]. However, in other organisms, such as *Thermoplasma acidophilum*, there are other MEV variants that use mevalonate 3-phosphate intermediates [69].

##### The Mevalonate Pathway in Parasitic Protozoa

Several *Leishmania* species and *Trypanosoma* possess enzymes of the mevalonate pathway. Experiments carried out by Low et al. showed that *T. brucei* bloodstream forms synthesize 1–12 isoprene-unit dolichols, all of which were metabolically labeled by [^3^H] mevalonate. Similarly, *Leishmania tropica* and *T. brucei* synthesize the polyprenylated side chain of UQ via the MEV pathway [70,71,72]. The routes that allow the synthesis of isoprenoids in *E. histolytica* have not yet been identified; however, in *G. lamblia*, one of the most closely related protozoans according to taxonomic group, the effects of inhibitors of the synthesis have been previously studied [106]. It was shown that the most widely used MVA pathway inhibitors for clinical applications, namely, mevastatin (also known as compactin or ML-236B) and mevinolin, affected trophozoite growth, an effect that could be reversed by the addition of mevalonate in the culture medium [106]. Even *Blastocystis*, *Acanthamoeba* and some apicomplexan parasites, including *Toxoplasma*, *Cryptosporidium* and *Plasmodium*, showed susceptibility to statins and apparent enzymatic capacity to metabolize MVA in protein extracts [76,77,84,107,108,109].

#### 3.2.2. The Methylerythritol 4-Phosphate Pathway

##### Generalities about the Methylerythritol 4-Phosphate Pathway

The MEP pathway begins with 1-deoxy-d-xylulose 5-phosphate synthase (DXS), a thiamine-dependent enzyme that condenses a glyceraldehyde 3-phosphate molecule with a pyruvate molecule to form 1-deoxy-d-xylulose-5-phosphate (DOXP). Then, the enzyme 1-deoxy-d-xylulose 5-phosphate reductase isomerase (DXR) forms 2-C-methyl-d-erythrol-4-phosphate (MEP). Fosmidomycin was initially identified as a specific inhibitor of the DXR enzyme but is currently known to also have inhibitory activity against other MEP enzymes [110]. In the next step, the enzyme 2-C-methyl-d-erythro-4-phosphate cytidine transferase (MCT) condenses MEP with cytidine triphosphate (CTP). The obtained product is 4-(cytidine-5-diphosphate)-2C-methyl-d-erythritol (CDP-ME). The enzyme 4-(cytidine-5-diphosphate)-2C-methyl-d-erythro-kinase phosphorylase, using an ATP molecule, catalyzes the conversion of CDP-ME to 2-phospho-4 (cytidine 5′-diphosphate)-2-methyl-d-erythritol (CDP-MEP). Subsequently, the enzyme 2C-methyl-d-erythritol-2,4-cyclodiphosphate synthase (MCS) transforms CDP-MEP to 2C-methyl-d-erythritol-2,4-cyclodiphosphate (MEcPP). Then, the enzyme 4-hydroxy-3-methylbut-2-enyl-diphosphate synthase (GcpE) reduces MEcPP to 1-hydroxy-2-methyl-2-(*E*)-butenyl 4-diphosphate (HMBPP). Finally, HMBPP is transformed to IPP/DMAPP by the enzyme hydroxymethylbutenyl diphosphate reductase (LytB) [96,111].

##### The Methylerythritol 4-Phosphate Pathway in Parasitic Protozoa

The growth inhibitory effect of fosmidomycin on *Plasmodium* spp., *Babesia* spp. and *Theileria* spp. and the detection of DOXP reductoisomerase activity in *P. falciparum* protein extracts were the first indicators of the existence of the MEP pathway in apicomplexan parasites [79,80,85,112,113]. MEP enzymes and intermediates were identified in *P. falciparum* [80,85], in addition to the inhibitory effects of fosmidomycin in some intermediates and final products of isoprenoid biosynthesis [85,86]. In other apicomplexan parasites such as *T. gondii* and *Eimeria tenella* the MEP pathway has been identified as an isoprenoid source [10,79,80,85,112,113,114,115]. However, fosmidomycin seems to have little effect on the viability of *Toxoplasma*, *Theileria* and *Eimeria* [10,86] and fosmidomycin recrudescence phenomena appear to be frequent in malaria [112]. Therefore, MEP pathway role and fosmidomycin resistance phenomena had been extensively studied in *Plasmodium* and *Toxoplasma* rather than other apicomplexan parasites.

In *P. falciparum* it was demonstrated that the antiparasitic activity of fosmidomycin as well as of some antibiotics that cause loss of the apicoplast can be grown indefinitely in asexual blood stage culture if exogenous IPP is proportioned. The authors suggested that isoprenoid precursor biosynthesis is the only essential function of the organelle at least during intraerythrocytic stages [79]. Also, in *Plasmodium*, Fosmidomycin-resistant *P. falciparum* parasites showed loss of PF3D7_1033400. PF3D7_1033400 encodes a homologue of haloacid dehalogenase-like sugar phosphatases (PfHAD1) which dephosphorylates a variety of sugar phosphates, including glycolytic intermediates [112]. Parasites lacking PfHAD1 have increased MEP pathway metabolites and therefore it was suggested to control substrate availability to the MEP pathway.

Focussing in *T. gondi*, knockout of both DOXP reductoisomerase and LytB demonstrated to be lethal for the parasite. However, it was previously commented that fosmidomycin seems to have little effect on the viability of *Toxoplasma* [115]. Furthermore, intracellular stages of *T. gondii* mutants that lack the enzyme farnesyl diphosphate/geranylgeranyl diphosphate synthase (TgFPP/TgGGPPS) have only a mild growth phenotype and an isoprenoid composition similar to wild type parasites However, extracellular parasites variabilities were clearly affected by the loss of the enzyme. Furthermore, the synergy between statin treatments and pharmacological or genetic interference with the parasite isoprenoid pathway. Due this, it was performed in vivo experiments which demonstrated to be possible to cure mice with atorvastatin (a statin) from a lethal infection with the TgFPPs mutants. These results suggest that *Toxoplasma* salvages FPP and GGPP from the vertebrate host and so it relies on both endogenous and human isoprenoid biosynthesis [10]. Finally, some authors suggested that fosmidomycin resistance in several pathogenic organisms to be due to the poor ability of the compound to penetrate the membrane [115,116,117]. In this sense, in *T. gondii* the parasite plasma membrane is a critical barrier to drug uptake [115]. Furthermore, mice infected with transgenic parasites expressing a bacterial transporter protein related to fosmidomycin efficacy can be cured from *Toxoplasma* by using this compound.

#### 3.2.3. Isoprenic Chains Elongation

##### Generalities about Isoprenic Chains Elongation

Regardless of the isoprenoid biosynthetic pathway, IPP and DMAPP can be transformed to their respective isomers by the action of an IPP isomerase [96]. The isomerization of IPP and DMAPP and subsequent elongation of isoprenic chains seems to occur in most living organisms [96,118] (Figure 1). The simplest elongation occurs by condensation of the IPP molecule and DMAPP to form geranyl pyrophosphate (GPP) by the enzyme GGP synthase. Subsequently, GPP can be condensed with another IPP molecule to form FPP by the enzyme FPP synthase. If FPP is condensed with IPP once again, geranylgeranyl pyrophosphate (GGPP) is formed, a reaction catalyzed by the enzyme GGPP synthase. This elongation process can continue for the formation of chains of different sizes [119,120,121,122]. For example, chains of nine isoprenic units (solanesyl pyrophosphate) can be formed for UQ-9 biosynthesis.

##### Isoprenic Chains Elongation in Parasitic Protozoa

Isoprenoid metabolism-based chemotherapeutic strategies against parasitic protozoa include not only the MEP and MVA pathways but also the elongation of isoprene chains. Risedronate and its analogs are compounds that are used in human and veterinary clinics for prevention and treatment of osteoporosis, but in trypanosomatids and apicomplexan parasites, these compounds have been shown to be effective as FPP/GGPP synthase inhibitors and affect the calciosomes [123,124,125,126,127,128]. Several terpenes of essential oils, such as nerolidol or farnesol, appear to have growth inhibitory activity in different species, including apicomplexan and trypanosomatid parasites [129,130,131,132,133]. Some authors have suggested that these effects are due to the structural similarity of these compounds to prenyltransferase/synthase substrates, with which the abovementioned terpenes could compete [128,129,130,131,132]. Moreover, farnesol is also an MVA pathway inhibitor in some organisms [134].

##### Parasite-Specific Isoprenoids: Phytyl Pyrophosphate

As seen, the isoprenoid elongation process can produce chains of different sizes but there is also an isoprenic chain with a different degree of saturation, the phytyl-pyrophosphate [123] (Figure 4). Phytyl-pyrophosphate is produced by a reduction of GGPP by geranylgeranyl diphosphate/geranylgeranyl-bacteriochlorophyllide reductase in plants which is not putative in parasites. This isoprenic moiety is typical of photosynthetic organisms and is constituent of tocopherol, PK, chlorophyll and phytilated proteins [123]. Phytylated quinones have already been discovered in *Leishmania* and *Plasmodium* organisms [54,64]. The incorporation of [^3^H]-GGPP in phytylated prenylquinone strongly suggests that both parasites possess an active pathway for phytol biosynthesis. Moreover, the incorporation of [^3^H]-phytol into phytylated quinones in both parasites suggests an active phytol phosphorylation process for the salvage of phytol, which is typical of some photosynthetic organisms [123]. Since phytol/phytyl-PP metabolism is absent in humans, last commented phytol-related pathways should be considered a great drug target that is still unexplored.

## 4. Specific Prenylquinones

Here, we will specifically discuss prenylquinones that are found in several human pathogenic protozoa as well as the therapeutic implications of these compounds (Table 3). To better understand these compounds, first, the well-known biosynthetic pathway, functions and distribution of prenylquinones in nature will be explained. The table shown below contains examples of prenylquinones in pathogenic protozoa that will be cited, discussed and referenced in this review. We show a scheme with most of the biosynthetic pathways for prenylquinone biosynthesis in nature except for the UQ biosynthetic pathway. The mechanism by which tocopherol and methyl-1,4-naphtoquinones are biosynthesized in pathogenic protozoa is poorly understood. Therefore, for these prenylquinones, all the well-characterized pathways in nature are shown in Figure 5.

### 4.1. Ubiquinone

(UQ) is the most common and well-studied prenylquinone in several organisms (including parasitic protozoa), so large amounts of data, such as the amounts of each UQ homolog at different parasitic stages, are available. These data are too specific for the scope of this review. For further specific and detailed data regarding UQ in parasitic protozoa, please refer to two comprehensive reviews published in 1994 [135,146].

#### 4.1.1. Ubiquinone Biosynthesis and Distribution

Ubiquinone is a prenylquinone composed of an isoprene chain attached at the carbon-2 of a 1,4-benzoquinone (Figure 6). UQ is a prenylquinone that probably originated from proteobacteria. There is a theory that proteobacteria gives rise to mitochondria via an endosymbiotic process with other organisms. Therefore, UQ can be found in α-, β- and γ-proteobacteria as well as in several eukaryotic subcellular compartments [45,147].

The aromatic precursor of UQ, p-hydroxybenzoate, is formed from chorismate via the shikimate pathway in most bacteria and plants, while in eukaryotes, p-hydroxybenzoate is provided by tyrosine [45]. The conversion of chorismate to p-hydroxybenzoate is catalyzed by the enzyme chorismate-pyruvate lyase, encoded by the ubiC gene [45]. In the biosynthetic pathway in most organisms, p-hydroxybenzoate is prenylated by the p-hydroxybenzoate polyprenyltransferase enzyme (UbiA/Coq2), followed by modification of the prenylquinone head group (Figure 7). These modifications are performed by Ubi/Coq, but the enzymatic reaction order can differ among species. However, in all organisms, decarboxylation, hydroxylation and SAM-dependent methylation occur.

Although p-hydroxybenzoate is the classic aromatic precursor for UQ biosynthesis, a recent study in yeast and *E. coli* showed complete incorporation of [phenyl-^13^C_6_] aminobenzoic acid (pABA) into the prenylquinone [45]. This new pathway has not been elucidated to date because, in all organisms in which [phenyl-^13^C_6_] pABA incorporation into UQ occurs, the intermediate that undergoes deamination remains unknown. This discovery suggests that the mechanisms of action of sulfonamide and other classic antifolates could be involved with inhibition of UQ biosynthesis [148]. Whether pABA is the UQ aromatic precursor in other organisms has not been studied to date [148,149,150,151].

The final biosynthetic enzymes in eukaryotes appear to be located on the membrane, suggesting that after the biosynthesis of UQ, there exists some type of intracellular transporter [152,153,154]. The main function of UQ is electron transport between complexes I or II and complex III of the mammalian internal mitochondrial membrane. However, many other enzymes that modify the redox state of UQ are known. During photosynthesis, UQ can also act on the PSII of organisms such as *R. rubrum* [45]. Finally, among other functions of UQ, antioxidant activity, inhibition of lipid peroxidation and regeneration of tocopherol [45,57,58,59] have been described. The function of the existence of a predominant and specific UQ homolog in each species and/or cell compartment is unclear.

#### 4.1.2. Ubiquinone in Parasitic Protozoa

Ubiquinones are antioxidants that are involved in mitochondrial bioenergetics activities in protozoa that use oxidative phosphorylation [136,155]. Ubiquinones are also biosynthesized by parasites that live under low oxygen tension and do not possess mitochondria or possess only a remnant [135]. As explained above, it is suspected that anaerobic protozoa such as *Giardia* may have suffered an endosymbiosis event with a mitochondria ancestor [38]. Moreover, these parasites do not seem to possess the mitochondrial complex II (succinate dehydrogenase), complex III (ubiquinol-cytochrome C reductase) and complex IV (cytochrome c oxidase) [156], indicating the absence of conventional mitochondrial electronic transport as an energetic source. However, in 1994, UQ-9 was detected at 5 to 50 fold lower amounts in *E. histolytica* and 15 to 350 fold lower amounts in *G. intestinalis* than in aerobic organisms [135]. Small amounts of UQ-10 were also detected in *Trichomonas fetus* [135]. The eventual functions of these prenylquinones in the abovementioned parasites remain poorly understood.

Ubiquinones clearly functions as a mitochondrial electron carrier in aerobic parasites such as trypanosomatids. Species of *Leishmania* present different homologs of UQs, with lateral chains varying in length from 7 to 10 isoprenic units, with UQ-9 being the predominant homolog. Promastigote forms of *Leishmania amazonensis* presented only the homolog UQ-9, whereas amastigote forms presented the UQs with 7, 8, and 9 isoprene units [135]. The differences in respiratory capacity among the forms should be the factor responsible for this divergence in the UQ profile: it is known that promastigotes present well-developed mitochondria in comparison to the degenerate mitochondria observed in the amastigote stage (vertebrate intracellular stage) [157]. *Leishmania donovani* promastigotes have mostly UQ-9, but UQ-8 and UQ-10 are also detectable. In addition, promastigotes of *Leishmania major* have UQ-9 and UQ-8 in a minor proportion [135]. Unlike the promastigotes of *L donovani* and *L. major*, the promastigotes of *Leishmania pifanoi* contained equivalent concentrations of UQ-9 and UQ-10, but UQs containing 7 and 8 isoprenic units in the lateral chain were also detected in *L. amazonensis* promastigotes [135]. The total UQ concentrations in *L. donovani* promastigotes and promastigote and amastigote forms of *L. amazonensis* were substantially higher than those in some aerobically grown yeasts.

Among trypansomatids, Ferella et al. in 2006 concluded that epimastigote forms of *T. cruzi* synthesize mainly UQ-9, which stimulated further studies on the importance of the trypanosomal enzyme solanesyl-diphosphate synthase (TcSPPS) [137]. Furthermore, the UQ content of procyclic forms of *T. brucei* was studied by Clarkson et al. in 1979 and by Low et al. in 1991. In both cases, UQ-9 was shown to be synthetized by this parasite [72,136].

The asexual stages of *P. falciparum, Plasmodium lophurae* and *Plasmodium knowlesi* have an active and essential pathway for UQ-7,8,9 homologs [146]. In other *Plasmodium* species, during asexual stages, some UQ homologs were not detected. For example, UQ-9 was not detected in *Plasmodium berghei* and neither was UQ-7 in *Plasmodium cynomolgi*. Surprisingly, in *P. falciparum*, the biosynthetic pathways of different UQ homologs showed selectivity for radiolabeled isoprenic donors (IPP, FPP, GGPP) [138], which was suggested to be due to prenyltransferase/prenylsynthase selectivity for different isoprenic moieties [138]. Similar to trypanosomatids, the biological role of different UQ homologs in *Plasmodium* is poorly understood, but some authors have suggested that these compounds are involved in different types of oxidative stress [138]. Different homologs could be formed due to the promiscuity of UbiA for differently sized isoprenic chains or by a step-by-step UQ lateral chain formation process that has been previously suggested to occur in the fungus *Pneumocystis carinii*. The fungus may produce UQ-10 from UQ-7 by repetitively adding 5-carbon isoprenic moieties to the lateral chain [158,159], so UQ-7,8,9 can be considered intermediates for UQ-10 biosynthesis. This UQ biosynthetic pathway malaria parasites has been already investigated by confirming [^14^C] 4-hidroxybenzoate and radiolabeled isoprenic moieties incorporation into UQ [54,140]. However, pABA could also be an alternative UQ aromatic precursor in apicomplexan parasites since most of them form pABA for folate biosynthesis [149]. Furthermore, for both pABA and 4-hydroxybenzoate in *Plasmodium gallinaceum*, in vivo experiments in chicks suggested that exogenous incorporation from the host is essential for intraerythrocytic stages [149].

The good therapeutic effects and antiparasitic activities of hydroxynaphthoquinones have been previously demonstrated [160]. The best known compound among this class of compounds is atovaquone (2-[-trans-4-(chlorophenyl)-cyclohexyll-3-hydroxy-1,4-naphthoquinone]), an antiparasitic agent that is specifically used to treat malaria, toxoplasmosis, babesiosis and pneumocystis pneumonia [161,162]. The drug binds to the UQH_2_-binding site in the *bc1* complex of several parasites with high affinity compared to mammalian mitochondria [135,163], preventing mitochondrial ATP synthesis and maintaining a UQ pool [164].

In the treatment of malaria, the effects of atovaquone on parasite viability are not caused by a decrease in ATP synthesis. In *Plasmodium*, the lethal effects seem to be depletion of the UQ required for the activity of the enzyme dihydroorotate dehydrogenase (DHODH), which is necessary for pyrimidine biosynthesis [135,165]. In clinics, atovaquone efficacy is improved by combination with proguanil, a folate biosynthesis inhibitor. Proguanil is a prodrug that is metabolized to cycloguanil, which in turn inhibits the enzyme dihydrofolatereductase (DHFR) [166]. However, the synergic mechanism seems to be independent of DHFR inhibition [165,167]. Furthermore, proguanil (but not cycloguanil) has been demonstrated to increase the ability of atovaquone to collapse the mitochondrial membrane potential (ΔΨ m) via a poorly understood mechanism [165,167].

Unfortunately, in apicomplexan and trypanosomatid parasites, atovaquone resistance has been previously reported, even with pharmaceutical combinations containing other antiparasitic drugs [168,169,170]. In *P. falciparum* and *T. gondii*, this resistance is thought to be caused by mutations in the mitochondrial complex III UQ-binding site [168,169], while in *Leishmania infantum* promastigotes, some studies linked the phenomenon with changes in the membrane lipidic composition [170]. Considering that UQ analogs that are less apolar than atovaquone can rescue the toxic effects of atovaquone in *Plasmodium* and the parasitic ability to easily modify UQ biosynthesis [171], it would be interesting to also explore UQ biosynthesis as a mechanism involved in atovaquone resistance. It still has not been studied whether atovaquone resistance could involve UQ biosynthesis or the proportion between the different homologs. The only studies that have focused on the effects of atovaquone on UQ biosynthesis were performed in the fungus *Pneumocystis carinii*, which demonstrated that atovaquone inhibits UQ biosynthesis at 10 nM, exhibiting stimulation at 0.2 mM and inhibition again at 1 mM [158]. These surprising phenomena could be associated with the previously proposed unconventional UQ biosynthetic pathway [158,159]. The effects of atovaquone on UQ biosynthesis/redox state remain to be demonstrated in pathogenic protozoa.

### 4.2. Vitamin K

#### 4.2.1. Biosynthesis and Distribution of Vitamin K

Natural occurring methyl-1,4-naphtoquinones include PK (or vitamin K1) and MQs (including MQ-4, vitamin K2). Some of these substances are considered vitamins that are mostly exogenously obtained and are involved in several essential physiological processes in animals [45]. Naturally, these compounds are derived from the diet or the intestinal microbiota [172], are used as cofactors for γ-glutamyl carboxylase among other enzymes [45,173] and can be “redox-recycled” by the epoxide reductase vitamin K [173]. Although the chemical structures of PK and MQs are similar, the functions, origins and distribution of these compounds in nature are different (Figure 1). Reduced MQs rapidly become oxidized and inefficient in rich oxygen-saturated environments, and these compounds appear to have originated in the reducing atmosphere before the appearance of oxygenic photosynthesis [174]. The isoprenoid side chains of MQs can differ in the number of isoprene units in the lateral chain and in the degree of saturation [147]. Modified MQs such as the proteobacterial TpQ biosynthesized by *Thermoplasma* [175] have also been identified and are discussed in detail in Section 4.3.1.

Several MQs can be found in archaea [175,176,177] and bacteria [147]. The different MQs are mainly involved in respiratory and photosynthetic processes in prokaryotes but are also found in diatoms and red algae [45,51,178]. Inhibitors of MQ biosynthesis are also of clinical interest with regard to the pathogenic actinomycetes and *Corynebacterium* species [55,179]. For example, MQ and its sulfated derivate produced by *Mycobacterium tuberculosis* are believed to act in both respiration and in attenuation of the host immune response [180]. Phylloquinone appears to have originated later than MQs in Earth’s history, but the structure of PK differs from that of MQs only by the phytol chain. The biosynthesis of PK in photosynthetic organisms capable of performing oxygenated photosynthesis can be detected. PK essentially participates in oxygenated photosynthesis and is mainly located in chloroplasts linked to photosystem I [181,182]. It is believed that PK could also be involved in (ROS) production under stress and in the resistance to pathogenic attack [183]. Some authors have suggested that there are PK-mediated electron transport chains in plant plasma membranes [183].

There is only one fully characterized pathway for MQ biosynthesis, and another remains poorly understood. The first pathway forms the head group of prenylquinone from chorismate via the activity of several enzymes: MenF (isochorismate synthase), MenD (2-succinyl-5-enolpyruvyl-6-hydroxy-3-cyclohexene-1-carboxylate synthase), MenH (2-succinyl-6-hydroxy-2,4-cyclohexadiene-1-carboxylate synthase), MenC (succinyl benzoate synthase), MenE (succinyl benzoic CoA synthase), MenB (1,4-dihydroxy-2-naphthoyl-CoA synthase) and DHNA-CoA thioesterase [184]. Finally, the head group is prenylated by a DHNA prenyltransferase, and the naphthoate is methylated. The second pathway, known as the futalosine pathway, diverges from the classic pathway until the formation of DHNA [185]. The pathway differs from the classic pathway fundamentally by the condensation of a chorismate molecule, an inosine and a two-carbon molecule that remains unidentified for the futalosine pathway [186]. Phylloquinone is biosynthesized by a pathway that is analogous to the previously described classic MQ biosynthetic pathway. The major difference between the pathway for vitamin K1 biosynthesis and the fully characterized pathway for vitamin K2 biosynthesis is the isoprenic chain condensed to the aromatic head group [45,187].

#### 4.2.2. Menaquinone in Parasitic Protozoa

In *P. falciparum*, MQ-4 biosynthesis has been characterized by Tonhosolo et al. [60], but no putative enzymes have been identified for the biosynthesis of this compound. However, several MQs are also found in diatoms and red algae, the supposed endosymbiont apicomplexan parasitic ancestors so this could be the origin of MQ in *Plasmodium* [45,51,178].

Tonhosolo et al. [60] performed metabolic labeling experiments using radiolabeled GGPP, which did not elucidate whether MQ-4 is biosynthesized from chorismate or is just a product of isoprenic chain modification of host-incorporated vitamins [188]. Furthermore, the functions of MQ-4 in *Plasmodium* appear to be complex and are poorly studied. NADH type II dehydrogenase that was chemically isolated in vitro from *P. falciparum* extracts or expressed in bacteria was shown to be able to use MQ-4 and menadione (vitamin K aromatic head group), respectively [60,61]. It was also observed that the UQ/MQ ratio was apparently inverted when the culture was maintained under anaerobic/microaerophilic conditions. RO 48-8071 (4-bromophenyl [2-fluoro-4-[[6-(methyl-2-propenylamino)hexyl]oxy]phenyl]-methanone), an inhibitor of MQ biosynthesis, apparently produced an increase in the UQ pool while decreasing MQ biosynthesis [60].

In trypanosomatids, the biosynthesis of vitamin K has not been characterized. However, several *Trypanosoma* and *Leishmania* species have putative sequences for vitamin K epoxide reductase (VKOR) enzymes (*T. cruzi*, UniProt: Q4DT49; *Leishmania (L.) donovani*, UniProt: E9BJE1). Based on these findings, it would be interesting to explore vitamin K biosynthesis or the incorporation of vitamin K from external environments for trypanosomatid metabolic functions.

### 4.3. Tocopherol

#### 4.3.1. Biosynthesis and Distribution of Tocopherol

Also known as vitamin E, tocopherol contains a chromanol as an aromatic head group bound to a phytyl chain (Figure 8). Several forms of tocopherol have been described as intermediates of α-tocopherol biosynthesis (γ, β, δ-tocopherol), and the biosynthesis of this compound can differ greatly depending on the organism [189]. Tocopherols are mainly found in the thylakoids of chloroplasts of photosynthetic organisms, but the biosynthesis of tocopherols was also detected in *P. falciparum* and *Leishmania L. amazonensis* [64,190]. These compounds have several functions; however, the most well studied function is protection against lipid peroxidation [191,192,193]. Other studies indicate that in photosynthetic organisms, tocopherol specifically interacts with photosystem II (PSII) components and cytochromes [194,195]. In humans, α-TQ appears to play an important role as a hydrogen acceptor for fatty acid desaturation [196]. Moreover, the irreversible reduction of α-TQ in alpha-tocopheryl quinone by b-cytochromes in the bc(1) complex suggests that α-TQ in mitochondrial membranes causes downregulation of respiratory activities [197]. In most photosynthetic organisms, the biosynthesis of tocopherol, PQ, PQ derivatives (for example PQ-B and PQ-C) and plastochromanol begins when chorismate is converted to 4HPP and consecutively to homogentisate by the enzyme HPPD [198]. The homogentisate is prenylated by the enzyme homogentisate phytyltransferase (HPT) and PQ by the enzyme homogentisate solanesyltransferase (HST) in plants. Finally, various enzymes, such as VT1, VT3 and VT4, catalyze different modifications of α-tocopherol or γ-tocopherol. However, an alternative route for PQ biosynthesis using 4-hydroxybenzoate as an aromatic donor has been discovered in cyanobacteria [45,198].

#### 4.3.2. Tocopherol in Parasitic Protozoa

Tocopherol biosynthesis was detected in asexual stages of *P. falciparum* and in *L. amazonensis* promastigotes. No specific enzymes involved in tocopherol biosynthesis were predicted by bioinformatics approaches in either parasite. However, the incorporation of [^3^H]-GGPP in phytylated prenylquinone strongly suggests that both parasites possess an active pathway for phytol biosynthesis. Moreover, the incorporation of [^3^H]phytol into tocopherol in *Leishmania* suggests an active phytol phosphorylation process for the salvage of phytol, which is typical for some photosynthetic organisms [123]. In this sense, prenylquinone biosynthesis is inhibited in both organisms by classic HPPD inhibitors: strongly by usnic acid in *P. falciparum* and *L. amazonensis* and weakly by nitisinone in *L. amazonensis*. Inhibition with usnic acid showed a dose-response effect on the growth of both parasites, causing an increase in ROS levels [64,190]. It is not known whether the major lethal effects of usnic acid could be due to inhibition of the biosynthesis of the product derived from homogentisate or tyrosine, as occurs in other organisms [199].

The effect of usnic acid on parasite viability and oxidative stress was reversed by the addition of exogenous α-tocopherol in *Plasmodium* and *Leishmania* [64,190]. Simultaneously, antiplasmodial oxidative stress-generating compounds such as chloroquine and cercosporin caused an increase in the α-tocopherolquinone/α-tocopherol ratio in *P. falciparum*, which is indicative of the antioxidant function [62]. However, notably, other authors have previously demonstrated that treatment with usnic acid generates oxidative stress in hepatic cells (which do not biosynthesize tocopherol), an affect that can be reversed by exogenous addition of tocopherol [200]. Consequently, the origin of the oxidative stress of usnic acid treatment in *Plasmodium* is unclear. Nevertheless, tocopherol biosynthesis in *P. falciparum* appears stimulated in vitro when the parasites are grown under 20% oxygen [190], which was also interpreted as a defense mechanism against oxidative stress. Although tocopherol biosynthesis has been previously demonstrated in *P. falciparum*, there remain many uncertainties regarding the mechanism of the biosynthetic process. Specifically, there are no putative sequences for tocopherol biosynthesis annotated in the *P. falciparum* genome. Furthermore, metabolic labeling using isoprenic precursors only proved the polyprenylation process. The aromatic biosynthesis or external acquisition remains unexplored. Even before the characterization of tocopherol biosynthesis in apicomplexan parasites, it was shown that the infected red blood cell membranes contained 5 fold more vitamin E and C than uninfected membranes. However, α-tocopherolquinone was not detectable in infected red blood cell membranes. Thus, some authors suggested that a parasite-mediated vitamin redox modification had occurred [201]. In vivo experiments demonstrated that tocopherol obtained from a host diet could be essential and stimulated parasitic development [201]. In this sense, vitamin E-deficient diets have been shown to be effective for disease control in vivo [202].

Studies focused on *T. cruzi* and *P. berghei* transmission have reported HPPD inhibition effects in hematophagous vectors [199,203]. Usnic acid has been shown to inhibit vectorial stages of *Plasmodium*, whereas nitisinone has been shown to be extremely toxic for several hematophagous vectors, even at clinically achievable blood concentrations. Some authors have suggested HPPD inhibitors as selective insecticides, but these inhibitors should also be considered to control the spread of hematophagous vector-transmitted diseases [199,203]. Tyrosine degradation has also been demonstrated to be essential in *Leishmania* parasites as an excellent drug target for further exploration [89].

In several plants and algae, the enzyme phytoene synthase provides phytoene, which is then desaturated by the enzyme phytoene desaturase (PDS) for carotenoid biosynthesis. In plants, PDS is dependent on PQ as a hydrogen acceptor, whereas some fungi and bacteria use another cofactor [204,205]. For this reason, herbicides such as norfluorazon compete with PQ in the PDS structure and effectively inhibit carotenoid biosynthesis [206]. Additionally, HPPD inhibition interferes with tocopherol and PQ biosynthesis and consecutively blocks carotenoid formation in plants. Simultaneously, carotenoid biosynthesis was demonstrated in *P. falciparum* to be inhibited by norfluorazon, a PDS enzyme inhibitor in plants and algae [204,207]. It has not been possible to identify in *P. falciparum* a putative PDS or a cofactor that could be used for these purposes. However, treatment of the parasite with norfluorazon apparently led to increased phytoene levels and inhibition of carotenoid biosynthesis [207]. These findings were considered to be proof of the PDS-specific inhibitory activity of norfluorazon [207]. Given this information, it remains unknown whether *Plasmodium* could use prenylquinones for the biosynthesis of carotenoids.

### 4.4. Thermoplasmaquinone and Chlorobiumquinone

#### 4.4.1. Thermoplasmaquinone and Chlorobiumquinone Distribution

TpQ-7 or methylmenaquinone-7 was initially isolated from *T. acidophilum* and subsequently from *Wolinella succinogenes* and *Bacteroides gracilis* [208]. ChQ was found in the green sulfur bacteria *Chlorobium thiosulphatophilum* and *Chloropseudomonas ethylicum* [209,210,211]. TpQ was discovered by Collins and Lamgworthy in 1983 in the acidophilic bacterium *Thermoplasma acidophilum* [212]. Nuclear magnetic resonance (NMR) studies revealed that the structure of this compound corresponds to 2-[5 or 8]-L-dimethyl-3-heptaprenyl-1,4-naphthoquinone, with a methyl group at the 5 or 8 position of the naphtoquinone portion [213]. On the other hand, ChQ together with MQ-7 were found in the green sulfur bacteria *C. thiosulphatophilum* and *C. ethylicum*, and the structures of these compounds correspond to 1-oxomenaquinone-7 [65,209,211]. No biochemical studies have clearly revealed how ChQ and TpQ are biosynthesized, but some authors suggest that these compounds are derivatives of MQ-7 [210]. The role of both prenylquinones remains unclear, although it is believed that these compounds could be involved in similar functions as MQ as well as in oxidative phosphorylation [209].

#### 4.4.2. Thermoplasmaquinone and Chlorobiumquinone in Parasitic Protozoa

Nandi et al. [66] in 2011 found TpQ-7 in trophozoite plasma membranes from *E. histolytica*, suggesting that this compound could be one of the main quinones involved in TPMET. The reduction of 1,2-naphthoquinone-4-sulphonic acid (NQSA), ferricyanide and α-lipoic acid (ALA) is evident, which suggests that TPMET redox chains are present in *E. histolytica*. Due to these quinone-mediated redox chains, it is possible to transfer electrons from the intracellular NADH or NADPH oxidation reactions to other compounds located in the extracellular space [67]. In *E. histolytica* amoebic liver abscesses, this redox chain could be essential for parasitic survival [67]. Similarly, in promastigotes of *L. donovani*, ChQ was found in fractions containing purified plasma membranes. In addition, in this case, prenylquinone was suggested to be involved in ALA, NQSA and ferricyanide reduction [65].

In both *E. histolytica* and *L. donovani*, the two prenylquinones can be destroyed by UV light irradiation, which leads to TPMET loss [65,67]. However, this TPMET loss can be recovered by exogenous addition of TpQ or ChQ analogs [65,67]. Both works discuss the phylogenetic origin of prenylquinone. First, *E. histolytica* most likely acquired genes responsible for MQ biosynthesis [65,67]. On the other hand, ChQ was detected in only *L. donovani* and green sulfur bacteria, which is consistent with the endosymbiosis theory for trypanosomatid ancestors [65]. Both studies focused on TPMET, so it remains unknown how prenylquinones are biosynthesized and whether these compounds could be acting on other metabolic points [65,67]. Regardless, the absence of TpQ-7 and ChQ in animals makes these compounds interesting subjects for the discovery of novel etiological treatments [65,67].

## 5. Summary of Drug Targets Related to Prenylquinone Biosynthesis and Functions

Prenylquinones are molecules involved in essential metabolic reactions in all organisms, for example, prevention of lipoperoxidation, mitochondrial respiratory chain or as enzymatic cofactors.

Surprisingly, only few studies involving prenylquinones as drug targets have been conducted so far, most of them about UQ-related metabolism [41,54]. The study of these pathways is interesting for the discovery of new drug targets. Moreover, the biosynthesis of prenylquinone precursors should be also considered as interesting metabolic targets for exploration.

An important fact is that the aromatic and isoprenic precursors of the prenylquinones were more studied then their specific biosynthesis pathways. In fact, biosynthesis inhibition studies have only been performed in vitro for tocopherol (in *L. amazonensis* and *Plasmodium falciparum*) and MQ (also in the apicomplexan parasite) [62,64]. Several compounds known to interfere on shikimate pathway and aminoacid degradation as well as the different pathways for isoprenoid biosynthesis or isoprenic chain elongation have been demonstrated to possess antiparasitic activity.

In regard to the aromatic precursor formation in parasitic protozoa, tyrosine degradation has been demonstrated to be essential in *Leishmania* and those enzymatic inhibitors of tyrosine degradation and tocopherol biosynthesis affects *Leishmania proliferation* and *P. falciparum* [62,63,64,89]. Unlike *Trypanosomatidae* species, *Apicomplexa* pathogens seems to require the shikimate pathway for the synthesis of aromatic precursors and corroborating this statement, glyphosate and shikimic acid analogs demonstrated effect against apicomplexan pathogens however there is an evidence that the last enzyme of shikimate pathway is not essential for the *P. berghei* viability [73].

Regarding to the isoprenic precursor formation in parasitic protozoa drug targets are concentrated in MEP and MVA pathways. First of all, it was previously demonstrated that *G. lamblia* is sensitive to the statins mevastatin and mevinolin, two classical inhibitors of the MVA pathway and for consequence the isoprenic precursors synthesis [69,106]. In the case of *Blastocystis*, *Acanthamoeba* and some apicomplexan parasites, including *Toxoplasma*, *Cryptosporidium* and *Plasmodium*, it was demonstrated the susceptibility to several other statins and there is an evidence of enzymatic capacity to metabolize MVA in *P. falciparum* extracts [77,81,84,106,107,108]. Curiously, there is no evidence for pathways responsible for isoprenic chain biosynthesis in *E. histolytica*.

After the identification of MEP pathway in apicomplexan parasites a vast number of researchers concentrated their efforts around the compound fosmidomycin and analogs. Fosmidomycin is a specific inhibitor of the DXR (1-deoxy-d-xylulose-5-phosphate reductoisomerase) enzyme [79,80,109,110,111,112,113]. Several studies demonstrated the effect of Fosmidomycin on *Plasmodium* spp., *Babesia* spp. and *Theileria* spp. proliferation, however fosmidomycin has a poor effect on *Toxoplasma*, *Theileria* and *Eimeria* viability although the MEP pathway is a proven isoprenoid source for apicomplexan parasites, [10,79,85,109,110,111,112].

The elongation of isoprenoid chain is also a promisor chemotherapeutic target against parasitic protozoa. For example, in trypanosomatids and apicomplexan parasites the risedronate and its analogs have shown to be effective FPP/GGPP synthase inhibitors [121,122,123,124,125,126]. Also, nerolidol or farnesol (terpenes of essential oils), appear to have growth inhibitory activity in different species, including apicomplexan and trypanosomatid parasites probably because of their structural similarity with the substrates of enlongation pathway enzymes [127,128,129,130,131]. Another important therapeutic target related to prenylquinones that can be cited is the effect of hydroxynaphthoquinones in parasitic protozoa [160]. The best representant of this class of compounds is atovaquone, used to treat malaria, toxoplasmosis and babesiosis [161,162]. Atovaquone affects the electron transport chain with high affinity, blocking the UQ pool reoxidation [163,164]. It is known that atovaquone efficacy is improved by the combination with proguanil (a folate biosynthesis inhibitor). Proguanil is a drug that is metabolized to cycloguanil, which in turn inhibits the enzyme dihydrofolatereductase (DHFR) [166]. However, the synergic mechanism seems to be independent of DHFR inhibition [165,167]. Furthermore, proguanil has been demonstrated to increase the ability of atovaquone to collapse the mitochondrial membrane potential (ΔΨ m), it is still a poorly understood mechanism [165,167].

It is important to recapitulate here that the apparent function of MQ-4 in *Plasmodium* was explored and the MQ biosynthesis inhibitor (RO 48-8071) affected the parasite proliferation and seems to increase in the UQ pool while decreasing MQ biosynthesis [60]. In the case of tocopherol biosynthesis, it could be verified an antiparasitic effect of Usnic acid in *Plasmodium falciparum* and *L. amazonensis* and Nitisinone in *L. amazonensis*, two classical inhibitors of HPPD [64,190].

## 6. Conclusions

Human parasitic protozoa cause a large number of diseases worldwide. For some of these diseases, no effective treatments to has been identified to date and drug resistance has been observed. For all these reasons, the discovery of new etiological treatments is necessary. Most likely due to the protozoa variability, uncertain phylogenetic origin, endosymbiotic events and evolutionary pressure for adaptation to adverse environments a surprising variety of prenylquinones can be found in these organisms. In this review were discussed prenylquinones that have been previously characterized in human pathogenic protozoa. UQ is the most widespread but also MQ, tocopherols, ChQ and t TpQ can be found *Amoebozoa*, *Apicomplexa*, and *Euglenozoa* philas. As seen in this review, all prenylquinones have a correlation with the endosymbiotic events that occur within these organism’s evolutionary history. These metabolites could be involved in essential physiological functions, for example, prevention of lipoperoxidation, participation in the mitochondrial respiratory chain, TPMET or as enzymatic cofactors. In order to better understand parasite physiology and to discover new drug targets it is recommended to continue the research on the protozoan prenylquinone profile, biosynthesis pathway and biological functions.

## Figures and Tables

**Figure 1 molecules-24-03721-f001:**
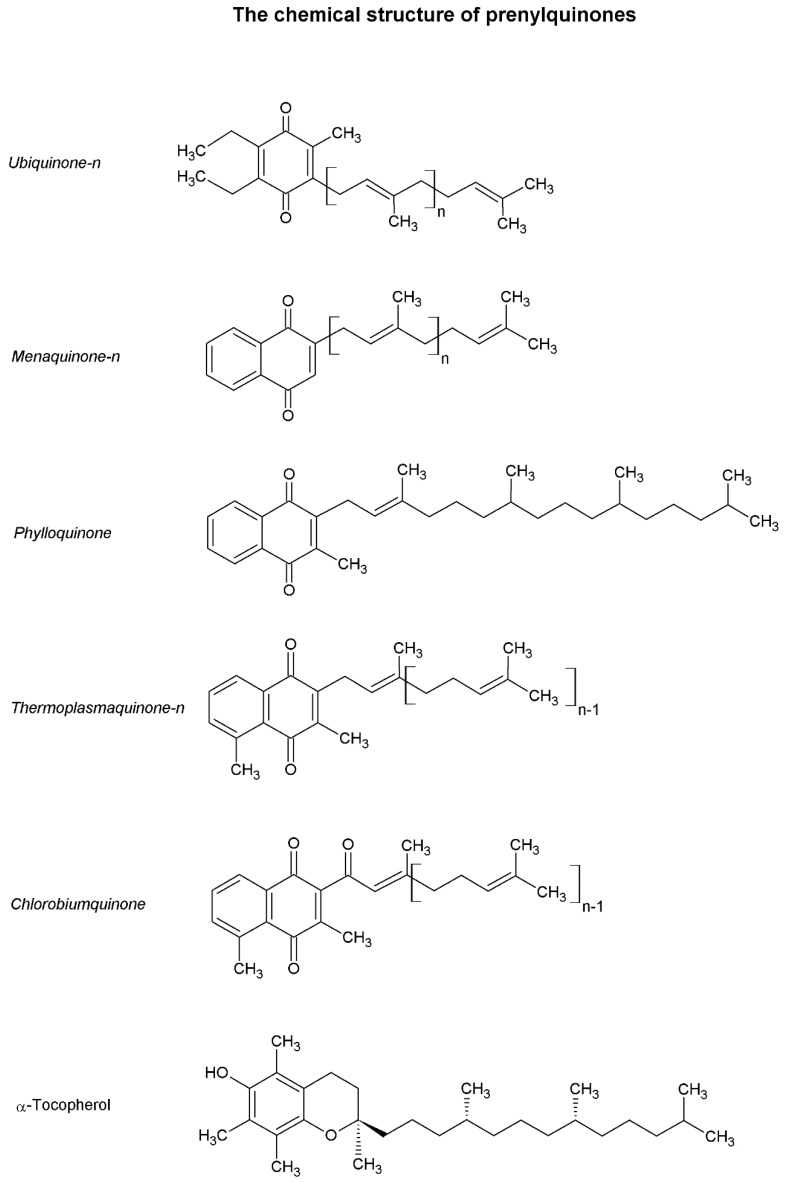
The chemical structures of prenylquinones. The figure shows the chemical structures of prenylquinones already characterized in pathogenic protozoa. *n* indicates the variable related to the isoprenic units contained on the prenylquinone side chain. Structures designed using ACD/ChemSketch, version 12, Advanced Chemistry Development, Inc., Toronto, ON, Canada.

**Figure 2 molecules-24-03721-f002:**
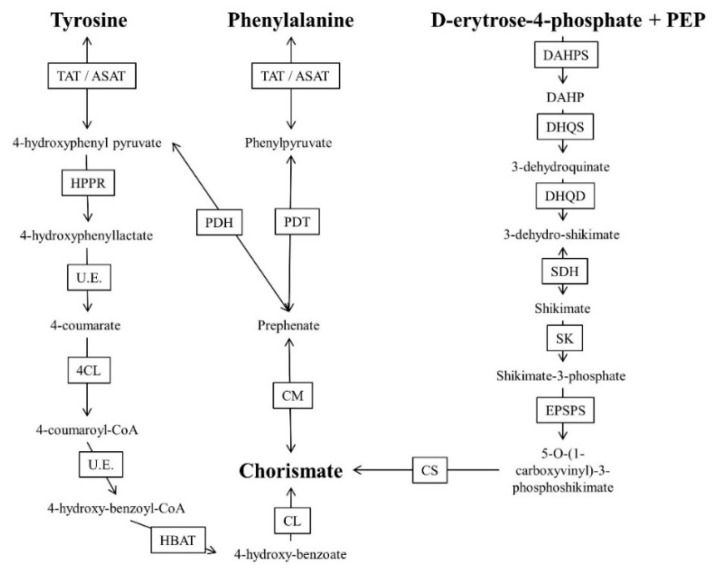
Scheme of chorismate synthesis from aromatic amino acids (tyrosine and phenylalanine) or the shikimate pathway. DAHP; PEP; TAT/ASAT (EC 2.6.1.5); HRPR (EC 1.1.1.237); U.E. (uncharacterized enzyme); 4CL (EC 6.2.1.12); HBAT (EC 3.1.2.23); CL (EC 4.1.3.40); PDH (EC 1.3.1.12/1.3.1.13); PDT (EC 4.2.1.51); CM (EC 5.4.99.5); DAHPS (EC 2.5.1.54); DHQS (EC 4.2.3.4); DHQD (EC 4.2.1.10); SDH (EC 1.1.1.25/1.1.5.8/1.1.1.282); SK (EC 2.7.1.71); EPSPS (EC 2.5.1.19); CS (EC 4.2.3.5). Based on information from Kyoto Encyclopedia of Genes and Genomes database [87].

**Figure 3 molecules-24-03721-f003:**
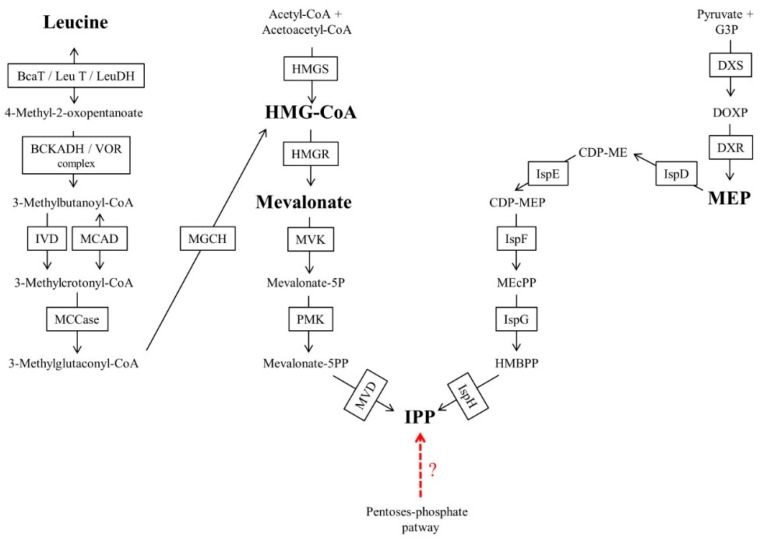
Scheme of isoprenoid biosynthesis up to isopentenyl pyrophosphate (IPP). The dotted, red arrow indicates the hypothetical IPP Biosynthesis Pathway previously mentioned in the text, starting from intermediate metabolites of the pentose phosphate pathway in *Synechocystis* sp. and a possibility that has not been investigated to date in any pathogenic protozoan [70]. BCAT/LeuT/LeuDH (EC 2.6.1.42/2.6.1.6/1.4.1.9); BCKADH/VOR complex (EC 1.2.4.4, 2.3.1.168, 1.8.1.4/1.2.7); IVD/ACADM (EC 1.3.8.4/1.3.8.7); MCCase (EC 6.4.1.4); MGCH (EC 4.2.1.18); HMGS (EC 2.3.3.10); HMGR (EC 1.1.1.34/1.1.1.88); MVK (EC 2.7.1.36); PMK (EC 2.7.4.2); MVD (EC 4.1.1.33); DXS (EC 2.2.1.7); DXR (EC 1.1.1.267); IspD (EC 2.7.7.60); IspE (EC 2.7.1.148); IspF (EC 4.6.1.12); IspG (EC 1.17.7.1/1.17.7.3); IspH (EC 1.17.7.4) Based on information from Kyoto Encyclopedia of Genes and Genomes database [87].

**Figure 4 molecules-24-03721-f004:**
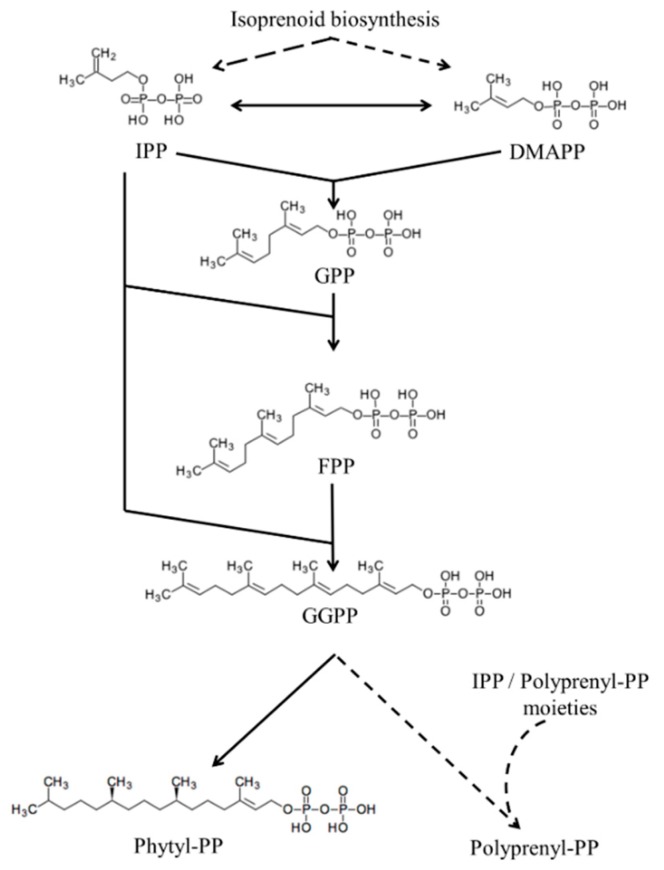
Isoprenic chain elongation/reduction. Isoprenic chain chemical structures and its elongation/reduction pathways. Continuous arrows indicate a single enzymatic step and discontinuous arrows indicate several enzymatic steps. Enzymatic steps are better described in the text. Structures sourced from Kyoto Encyclopedia of Genes and Genomes database [87].

**Figure 5 molecules-24-03721-f005:**
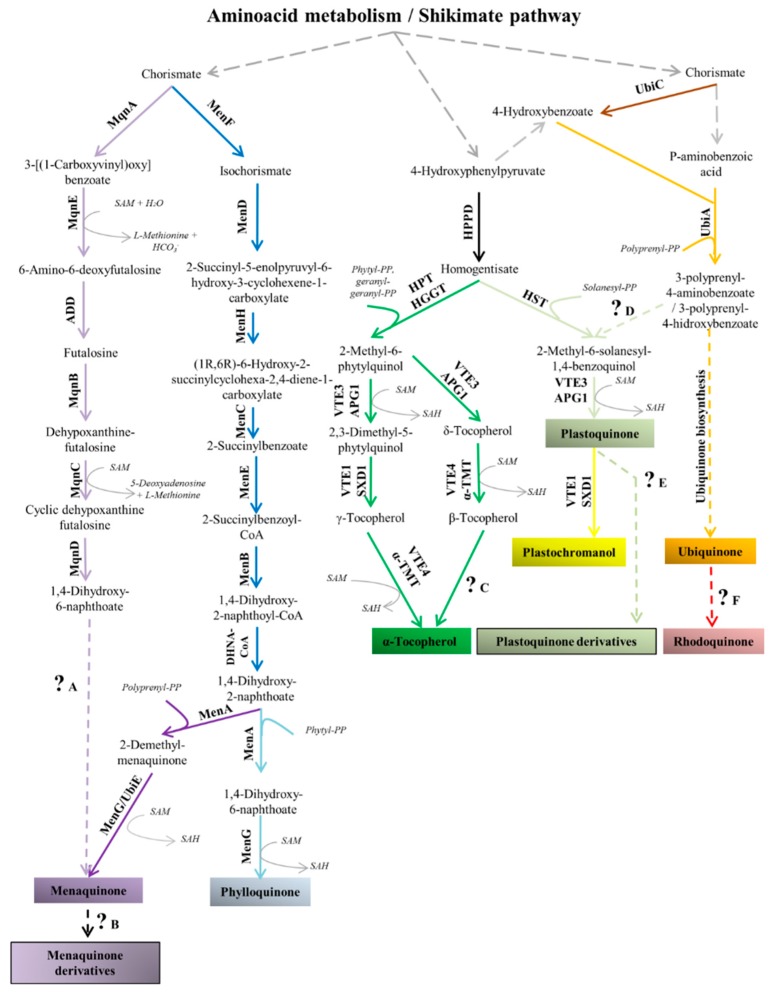
Prenylquinone biosynthesis pathways [45]. Some prenylquinones shown have not been found in pathogenic protozoa, but all are cited in this review. The specific pathway of ubiquinone biosynthesis (orange arrows) and the formation of aromatic precursors from amino acid metabolism/shikimate pathways (gray arrows) are simplified because these processes are better described in other figures in this review. The incorporation of isoprene chains and use of S-adenosyl-L-methionine (SAM) by enzymes are indicated. The SAM cofactor is necessary for methylation reactions to produce S-adenosyl homocysteine or for other biochemical processes. Continuous arrows indicate a single enzymatic step, and discontinuous arrows indicate several enzymatic steps, processes that remain poorly understood or processes that are better shown in other figures in this review. The different pathways are distinguishable by the colors of the arrows: Light purple. MQ biosynthesis by the futalosine alternative pathway; Dark blue. Homologous enzymatic steps that are common for PK or MQ biosynthesis by the classic pathway; Dark purple. Specific enzymatic steps for MQ biosynthesis by the classic pathway; Light blue. Enzymatic steps that are specific for PK biosynthesis; Black. Biosynthesis of homogentisate from 4-hydroxyphenylpyruvate; Dark green. Tocopherol biosynthesis [142]; Light green. Shared enzymatic steps for PQ and plastochromanol biosynthesis; Yellow. Plastochromanol biosynthesis [45]; Brown. 4-Hydroxybenzoate biosynthesis from chorismate; Orange. Ubiquinone biosynthesis; Red. Possible rhodoquinone biosynthesis pathway in *Rhodospirillum rubrum* [143]. The figure also indicates processes that remain poorly understood. ?A. The final steps of the futalosine alternative pathway [144]; ?B. Biosynthesis of MQ derivatives such as ChQ or sulfated MQs (among other examples) [45]; ?C. α-Tocopherol biosynthesis from β-tocopherol; ?D. 2-Methyl-6-solanesyl-1,4-benzoquinol biosynthesis from 4-hydroxy-3-polyprenylbenzoate in cyanobacteria [145]; ?E. The formation of some PQ derivatives (e.g., PQ-B, PQ-C) is controversial in the literature [45]; ?F. Rhodoquinone biosynthesis requires ubiquinone in *Rhodospirillum rubrum* [143], but the process remains poorly understood. Next to each arrow, the corresponding enzyme is cited. For biosynthesis of tocopherol, plastochromanol and PQ, different enzymes can perform the same enzymatic step depending on the organism [141]. Enzymes: MqnA (chorismate dehydratase, EC:4.2.1.151). MqnE. (aminodeoxyfutalosine synthase, EC:2.5.1.120). ADD (aminodeoxyfutalosine deaminase, EC:3.5.4.40). MqnB (futalosine hydrolase, EC:3.2.2.26). MqnC (cyclic dehypoxanthinyl futalosine synthase, EC:1.21.98.1). MqnD (EC 1.14.). MenF (EC 5.4.4.2). MenD (EC 2.2.1.9). MenH (EC 4.2.99.20). MenC (EC 4.2.1.113). MenE (EC 6.2.1.26). MenB (EC 4.1.3.36). DHNA-CoA (EC 3.1.2.28). MenA (EC 2.5.1.74). MenG/UbiE (demethylmenaquinone methyltransferase/2-methoxy-6-polyprenyl-1,4-benzoquinol methylase; EC:2.1.1.163/EC:2.1.1.201, respectively). HPPD (EC:1.13.11.27). HPT (homogentisate phytyltransferase/homogentisate geranylgeranyltransferase; EC:2.5.1.115/EC:2.5.1.116, respectively). VTE3/APG1 (EC 2.1.1.295). VTE1/DXD1 (EC 5.5.1.24). VTE4/α-TMT (EC 2.1.1.95). HST (EC 2.5.1.117). UbiC (EC 4.1.3.40). UbiA (EC 2.5.1.39). Based on information from the cited references in the text and the Kyoto Encyclopedia of Genes and Genomes database [87].

**Figure 6 molecules-24-03721-f006:**
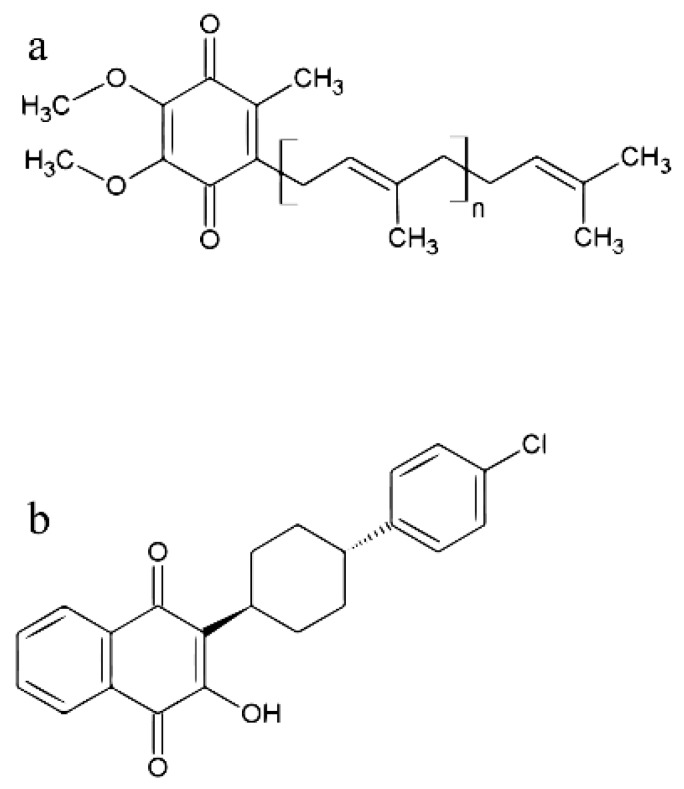
Chemical structures of ubiquinone and atovaquone. (**a**) Chemical structures of UQ, *n* indicates variable number of isoprenic units contained on the prenylquinone side chain. (**b**) The antiparasitic drug atovaquone.

**Figure 7 molecules-24-03721-f007:**
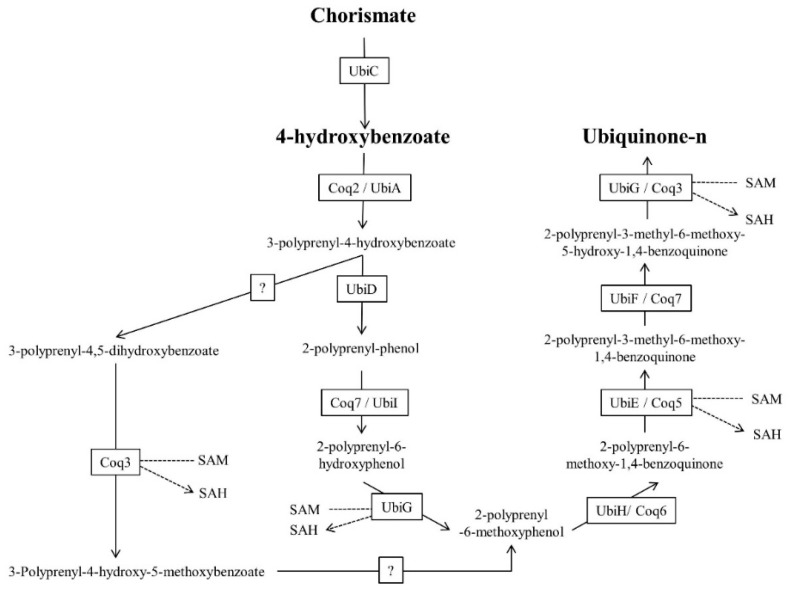
Scheme of ubiquinone biosynthesis. UQ biosynthesis from 4-hydroxybenzoate. In the figure depicting the metabolic pathways, the transformation of the cofactor SAM to S-adenosyl-l-homocysteine (SAH) for methylation is indicated. UbiC. Chorismate-pyruvate lyase, EC: 4.1.3.40; UbiA/Coq2. EC: 4-Hydroxybenzoate polyprenyltransferase, 2.5.1.39; UbiD. 4-Hydroxy-3-polyprenylbenzoate decarboxylase, EC: 4.1.1.98; Coq7/UbiI. 2-Polyprenylphenol 6-hydroxylase, EC: 1.14.13.240; UbiH/Coq6. UQ biosynthesis monooxygenase/2-octaprenyl-6-methoxyphenol hydroxylase, EC: 1.14.13.-; UbiE/Coq5. 2-Methoxy-6-polyprenyl-1,4-benzoquinol methylase, EC: 2.1.1.201; UbiF/Coq7. 3-Demethoxyubiquinol 3-hydroxylase, EC: 1.14.99.60; UbiG/Coq3. Polyprenyldihydroxybenzoate methyltransferase/3-demethylubiquinol, 3-*O*-methyltransferase/2-polyprenyl-6-hydroxyphenyl methylase/3-demethylubiquinone-9 3-methyltransferase, EC: 2.1.1.114/2.1.1.64/2.1.1.222/. Data from the Kyoto Encyclopedia of Genes and Genomes database [87].

**Figure 8 molecules-24-03721-f008:**
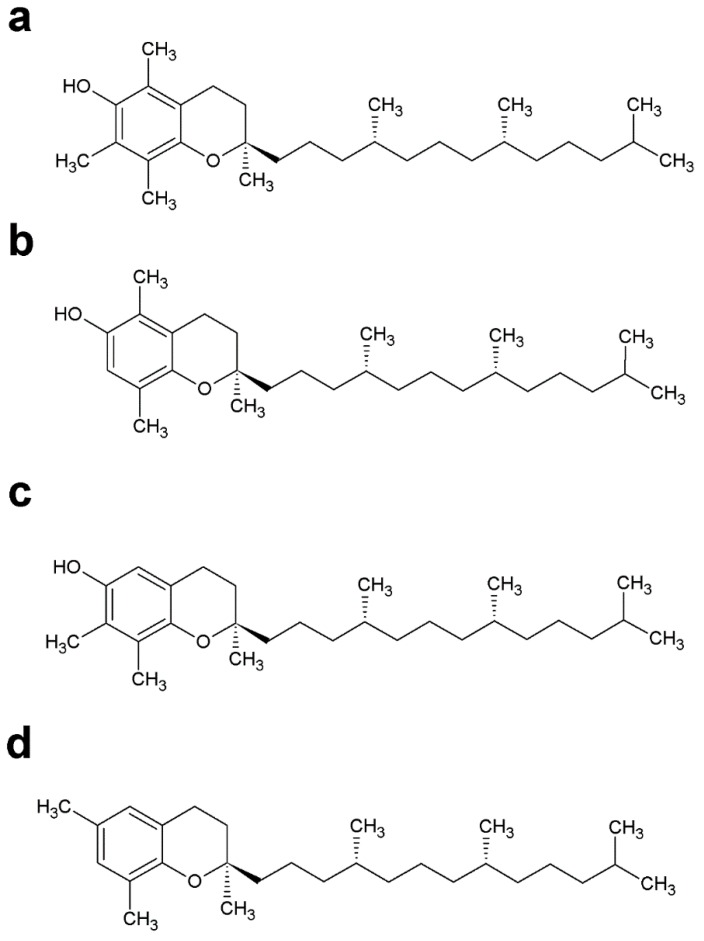
Chemical structures of tocopherols. (**a**) Chemical structures of alpha-tocopherol, (**b**) beta-tocopherol, (**c**) gamma-tocopherol and (**d**) delta-tocopherol.

**Table 1 molecules-24-03721-t001:** The table indicates the parasitic prenylquinone functions among protozoa. For more information, check the reference section. Grey squares contain examples of previously characterized prenylquinones from protozoa. White squares indicate still non-characterized prenylquinones.

Prenylquinone	Function		
	Antioxidant Defense	Mitochondrial Respiration	Trans-plasma Membrane Electron Transport
Ubiquinone	[45,57,58,59]	[50]	
Menaquinone		[60,61]	
Tocopherol	[62,63,64]		
Chlorobiumquinone			[65]
Thermoplasmaquinone			[66,67]
Rhodoquinone		[48]	

**Table 2 molecules-24-03721-t002:** Isoprenic/aromatic sources in pathogenic protozoa. The presence of aromatic amino acids degradation pathways (tyrosine and phenylalanine), the shikimate pathway and the isoprenoid source in pathogenic protozoa.

Organism	Aromatic Head Group Biosynthesis		Isoprenoid Side Chain Biosynthesis	
	Via Shikimate Pathway	Via Amino Acid Degradation	Mevalonate Pathway	Methylerythritol 4-Phosphate Pathway
*Leishmania spp.*	-	Indirect evidence [63]	Experimental evidence at protein level [68,69]	-
*Trypanosoma cruzi*	-	Protein predicted	Experimental evidence at protein level [70]	-
*Trypanosoma brucei*	-	Protein predicted	Experimental evidence at protein level [71,72]	-
*Plasmodium* spp.	Experimental evidence at protein level [73,74,75]	-	Indirect evidence [76,77]	Experimental evidence at protein level [78,79,80]
*Giardia intestinalis*	-	-	Indirect evidence [81]	-
*Trichomonas vaginalis*	-	-	Protein predicted *	-
*Toxoplasma gondii*	Proteins inferred from homology [73,82,83]	-	Indirect evidence [84]	Experimental evidence at protein level [20,85,86]

**Table 3 molecules-24-03721-t003:** Examples of the prenylquinone content in pathogenic protozoa. Table indicates in grey squares examples of previously characterized prenylquinones from pathogenic protozoa. White squares indicate still not characterized prenylquinones. All the prenylquinones are discussed in this review [64,72,135,136,137,138,139,140,141].

Organism	Parasitic Stage	UQ-8	UQ-9	UQ-10	ChQ	TpQ	MQ-4	PK	α-TC, γ-TC
*L. amazonensis*	amastigote								
promastigote								
*L. donovani*	promastigote								
*L. major*	promastigote								
*L. mexicana*	amastigote								
promastigote								
*T. cruzi*	epimastigote								
*T. brucei*	procyclic								
*P. falciparum*	intraerythrocytic								
*G. lambia*	trophozoite								
*E. histolytica*	trophozoite								
*Trichomonas foetus*	-

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
