# Peer review of "Prenylquinones in Human Parasitic Protozoa: Biosynthesis, Physiological Functions, and Potential as Chemotherapeutic Targets"

_molecules, 2019, doi:10.3390/molecules24203721_

Round 1
Reviewer 1 Report
Manuscript Number: Manuscript ID: molecules-578450
Title: Prenylquinones in human parasitic protozoa: 2 biosynthesis, physiological functions and potential as 3 chemotherapeutic targets
In my opinion, the study is of scientific relevance, it is interesting and should be published in International Journal of Molecular Sciences journal.
This review describe several prenylquinones that have been previously characterized in human pathogenic protozoa. Among all existing prenylquinones this review is focused on ubiquinone, menaquinone, tocopherols, chlorobiumquinone and thermoplasmaquinone. Authors also discuss the biosynthesis of prenylquinones.
Strong points of the paper: topicality of the subject, the social utility, especially human parasitic protozoa cause a large number of diseases worldwide. For some of these diseases there are no effective treatments to date and drug resistance has been observed. For all these reasons the discovery of new etiological treatments is necessary.
Author Response
As we already submitted the manuscript in the journal Molecules we consider that it will be submitted again in this journal
Reviewer 2 Report
The review article “Prenylquinones in human parasitic protozoa: biosynthesis, physiological functions and potential as chemotherapeutic targets” by Verdaguer et. al. is suitable for publication after addressing the following minor comments:
Authors should include another paragraph describing Prenylquinones’ potential as chemotherapeutic targets
Author Response
Reviewer 2
A paragraph describing Prenylquinones’ potential as chemotherapeutic targets was included.
Reviewer 3 Report
This review focused on the prenylquinones, a wide varieties of bioactive compounds involved in essential metabolic reactions in organisms, described their generalities and specifically provided detailed reviews in human parasitic protozoa, frequent diseases occurred in tropical and subtropical areas. Main biosysthesis pathways, physiological functions of these compounds and representative drugs existed were comprehensively discussed, including their targets and mechanism as well as drawbacks and resistance generated during evolution or adaptation, putting forward some potential therapeutic targets for future development and better control of these diseases. In my opinion, this is a quite comprehensive review in introducing the biosynthesis and their physiological functions of prenylquinones, providing a whole picture of these bioactive compounds in parasitic protozoa. However, I think some flaws should be well addressed and more information would be supplemented to be qualified for publication as listed below:
Major comments:
First of all, no doubt, this review provided quite substantial information regarding the biosysthesis and physiologic functions of prenylquinones in organisms as well as some representative ‘old’ drugs and related resistance already generated. Nevertheless, quite limited information regarding the new potential therapeutic targets and corresponding strategies or candidates were introduced in this review, unavoidably giving an impression of anticlimax. It would be great helpful if related contents could be summarized together here to give a more detailed picture of prenylquinones.Quinones, generally including naphthoquinones or benzoquinones, are organic compounds derived from aromatic compounds with conversion of -CH= into -C(=O)- and/or rearrangement of double bonds, but not all are composed of an aromatic ring, e.g., Ubiquinone-n (Line 123).
Line 165-167, NADH-rhodoquinone oxidoreductase and rhodoquinol-fumarate reductase are the exact target, but rhodoquinol-fumarate itself is substrate, not enzyme or enzymatic target.
What is the aim of putting two organisms without any evidences of characterized metabolic pathways in Table 2 – Entamoeba histolytica and Cryptosporidium spp? By the way, It is not quite clear that in Line 188-190, ‘Grey squares indicate previously characterized metabolic pathways from pathogenic protozoa and includes the information source. White squares indicate still not characterized pathways’. As I understood, Organisms in line 1, 3, 5, 7, 9 had metabolic pathways characterized before while those in line 2, 4, 6, 8 did not, correct? But there were not any pathways listed in line 6 & 9. What does this mean?
With respect to amino acid degradation, converting amino acid to 4-hydroxybenzoate, were there any references to support that the formation of 4- hydroxyphenylactate occurred via an (‘an’ before N /en/, or ‘a’ before NAD /næd/?) NAD+-dependent HPP reductase? In the third step, Ref 64 demonstrated the biosynthesis of 4-coumarate into p-hydroxybenzoate rather than the conversion of HPL into 4-coumarate. On the other hand, it was mentioned there the fifth step was still controversial but only one opinion was listed here from Loscher et al (Ref 64), which clearly elucidated the conversion of 4-coumarate into 4-hydroxybenzoate via a 4CL, ATP, CoA step followed by β-oxidation and enzymatic hydrolysis by thioesterase. Any other representative mechanism or hypothesis?
What might be the possible reasons for the failure of fosmidomycin in treating T. gondii and Eimeria tenella? And what should be paid extra attentions since these well-developed drugs were not able to be extended to similar species? Like, mentioned here, penetration issues or any others? And any solutions?
Line 668, what is the ‘bioinformatics analyses’ carried out here? Could this be better illustrated in this review to provide more details of several species
Line 706, it was mentioned here biosynthesis of both phycol-PP and tocopherol were absent in humans, which could be advantageous drug targets. Were any research results or updates regarding these available to solid this viewpoint or if there was candidate already under evaluation?
Minor comments:
Line 26, maybe come from; Line 49, samples; Line 85, due to this phenomena; Line 97 & 98, suggested; Line 115 & 694, incorrect α-Tocopherol structure; Line 189, include; Line 259-260, what ‘so the essentiality of pathway in the parasite under review’ meant? Line 313-314, . ‘The MVA pathway begins with the condensation of two acetyl-CoA molecules for HMG) formation’? Line 389, due to; Line 396, hyphen before ‘a’? Line 397, ‘typical of’? Line 398, ‘As it will better seen in chapter 3’? Line 478, gives; Line 516, could be involved; Line 577, most of them; Line 722, is; Line 732, stimulated;
Extensive grammar or spelling errors may exist besides listed above. Meanwhile the language should be more scientific and logical. Please check carefully and make proper editing.
Author Response
Answer to reviewer 3
First of all, no doubt, this review provided quite substantial information regarding the biosynthesis and physiologic functions of prenylquinones in organisms as well as some representative ‘old’ drugs and related resistance already generated. Nevertheless, quite limited information regarding the new potential therapeutic targets and corresponding strategies or candidates were introduced in this review, unavoidably giving an impression of anticlimax. It would be great helpful if related contents could be summarized together here to give a more detailed picture of prenylquinones.
Quinones, generally including naphthoquinones or benzoquinones, are organic compounds derived from aromatic compounds with conversion of -CH= into -C(=O)- and/or rearrangement of double bonds, but not all are composed of an aromatic ring, e.g., Ubiquinone-n (Line 123).
Line 165-167, NADH-rhodoquinone oxidoreductase and rhodoquinol-fumarate reductase are the exact target, but rhodoquinol-fumarate itself is substrate, not enzyme or enzymatic target.
The text was modified to refer to the enzyme and not substrate as indicated.
What is the aim of putting two organisms without any evidences of characterized metabolic pathways in Table 2 – Entamoeba histolytica and Cryptosporidium spp? By the way, It is not quite clear that in Line 188-190, ‘Grey squares indicate previously characterized metabolic pathways from pathogenic protozoa and includes the information source. White squares indicate still not characterized pathways’. As I understood, Organisms in line 1, 3, 5, 7, 9 had metabolic pathways characterized before while those in line 2, 4, 6, 8 did not, correct? But there were not any pathways listed in line 6 & 9. What does this mean?
Following the suggestions, Entamoeba histolytica and Cryptosporidium spp were removed from the Table 2. The legend and the table format were changed to better understanding.
With respect to amino acid degradation, converting amino acid to 4-hydroxybenzoate, were there any references to support that the formation of 4- hydroxyphenylactate occurred via an (‘an’ before N /en/, or ‘a’ before NAD /næd/?) NAD+-dependent HPP reductase?
We included a new reference to support this affirmation
In the third step, Ref 64 demonstrated the biosynthesis of 4-coumarate into p-hydroxybenzoate rather than the conversion of HPL into 4-coumarate. On the other hand, it was mentioned there the fifth step was still controversial but only one opinion was listed here from Loscher et al (Ref 64), which clearly elucidated the conversion of 4-coumarate into 4-hydroxybenzoate via a 4CL, ATP, CoA step followed by β-oxidation and enzymatic hydrolysis by thioesterase. Any other representative mechanism or hypothesis?
There is no other evidence in the literature to generate the mentioned controversy, so we removed that affirmation.
What might be the possible reasons for the failure of fosmidomycin in treating T. gondii and Eimeria tenella? And what should be paid extra attentions since these well-developed drugs were not able to be extended to similar species? Like, mentioned here, penetration issues or any others? And any solutions?
Clastre et al., 2007 exposed five hypotheses to the little effect of fosmidomycin in E. tenella and T. gondii: “(i) biotransformation of the drug to inactive metabolites, (ii) inability of the drug to cross the host cell and parasite membranes, (iii) efflux of the drug from the parasite as reported for fosmidomycin-resistant E. coli bacteria (Fujisaki et al., 1996), (iv) nature of the host cell type (red blood cells versus epithelial cells or fibroblasts)”
Line 668, what is the ‘bioinformatics analyses’ carried out here? Could this be better illustrated in this review to provide more details of several species
No ‘bioinformatics analyzes’ were performed, we only searched for putative sequences in the genome of the mentioned organisms, consequently that phrase was removed.
Line 706, it was mentioned here biosynthesis of both phycol-PP and tocopherol were absent in humans, which could be advantageous drug targets. Were any research results or updates regarding these available to solid this viewpoint or if there was candidate already under evaluation?
There are only two evidences of Phytol-PP/tocopherol biosynthesis in parasitic protozoa. There is no available drug inhibiting this metabolism and because of this we decided to remove the affirmation about “therapeutic target”, but still interesting to investigate this metabolism since is absent in human.
Minor comments:
Line 26, maybe come from; Line 49, samples; Line 85, due to this phenomena; Line 97 & 98, suggested; Line 115 & 694, incorrect α-Tocopherol structure; Line 189, include; Line 259-260, what ‘so the essentiality of pathway in the parasite under review’ meant? Line 313-314. ‘The MVA pathway begins with the condensation of two acetyl-CoA molecules for HMG) formation’? Line 389, due to; Line 396, hyphen before ‘a’? Line 397, ‘typical of’? Line 398, ‘As it will better seen in chapter 3’? Line 478, gives; Line 516, could be involved; Line 577, most of them; Line 722, is; Line 732, stimulated;
We considered all minor comments and we changed as indicated by the reviewer. The tocopherol structures in Figure 1 and 8 were corrected.
Round 2
Reviewer 3 Report
The revised manuscript answered my questions and concerns in some way. Necessary contents were also modified accordingly in the manuscript as well as in part 5 – potential drug targets. I recommended for publication after careful revision in language and grammar. Some editing work listed below were still required but not limited to:
The structure of tocopherol is still incorrect in Fig 1 and 8. It is -OH group on 6’ position of benzene ring, not -CH3. In Fig 1, the alpha-tocopherol is not correct as well as in Fig 8 (https://en.wikipedia.org/wiki/Tocopherol). On the other hand, the chiral conformation of beta-tocopherol in Fig 8 is missed (not specified on the long chain). Maybe typos but there is two exact the same Table 1. Similar issue in Table 3. Line 945, the bullet number is incorrect. Line 982, either ‘regarding’ or ‘in regard to’, not ‘regarding to’ Line 977, ‘then’? Line 1004 and 1009, maybe ‘precursor’ is a strategy or a target? Line 1008-9, ‘because their structural similarity with substrates of enlongation pathway enzymes’, no verb. Line 1019, ‘It is important…’. The tense consistency should be paid much more attention to as well as singular or plural form in one sentence.This manuscript is a resubmission of an earlier submission. The following is a list of the peer review reports and author responses from that submission.
Round 1
Reviewer 1 Report
It is very comprehensive study on prenylquinones in human parasitic protozoa. I have only suggestion to rewrite abstract part more concretely (to emphasize to the most specific and important facts), in the same way in the conclusion part, sum up all points of review article.
Author Response
The abstract and conclusions were modified as reviewer suggest.
Reviewer 2 Report
Line 22: several "the" --> "the" should be removed.
Author Response
In line 23 the modification was introduced
Reviewer 3 Report
The manuscript provides distribution and biosynthesis of some prenylquinones parasitic protozoa. I think it describes in details, however, it is too hard to read it through with lots of text. I would say that it is important for reader friendly manuscript to use Table, figure, and scheme appropriately. The authors might want to add table along with the topics of each chapter. Also, adding chemical structures in each figure 1-5 can help us understand what the authors are going to explain. Please keep in mind that chemists are in the majority of the readers of this journal. Examples for Table and Figure are as follows:
Cited from Infect Immun. 2017, 85(8), e00101-17.
Cited from Plant Methods 2011, 7, 23.
Author Response
We appreciate the criticisms made by the reviewer and introduce some modifications in figure 3. We do not add the chemical structures in figures 1-5 because they would make reading difficult. The chemical structures can be seen in different web pages if the reader is interested. The suggestion made by the reviewer is appropriate when the work has a theme addressed to the synthesis and in our case the review discusses biosynthetic pathways.
Round 2
Reviewer 3 Report
Problems still remains unsolved. Again, I recommend that the authors add new tables along with the topics of each chapter to improve readability. Examples have been already provided in my previous report.